# High-Hydrostatic-Pressure-Stabilized White Grape Pomace to Improve the Oxidative Stability of Dry-Cured Sausages (“Salchichón”)

**DOI:** 10.3390/foods13050687

**Published:** 2024-02-24

**Authors:** Ana Isabel Carrapiso, María Jesús Martín-Mateos, Matilde D’Arrigo, Jonathan Delgado-Adámez, Jorge Alexandre Saraiva, María Rosario Ramírez-Bernabé

**Affiliations:** 1Tecnología de Alimentos, Escuela de Ingenierías Agrarias, Universidad de Extremadura, 06006 Badajoz, Spain; acarrapi@unex.es; 2Instituto Tecnológico Agroalimentario (INTAEX), Centro de Investigaciones Científicas y Tecnológicas de Extremadura (CICYTEX), 06187 Badajoz, Spain; mariajesus.martinmat@juntaex.es (M.J.M.-M.); matilde.darrigo@juntaex.es (M.D.); jonathan.delgado@juntaex.es (J.D.-A.); 3Research Unit of Organic Chemistry, Natural and Agro-Food Products (QOPNA), Chemistry Department, Aveiro University, 3810-193 Aveiro, Portugal; jorgesaraiva@ua.pt

**Keywords:** grape pomace, wine by-product, dry-cured sausage, nitrite-free meat, antioxidant, oxidation, high hydrostatic pressure, HHP

## Abstract

White grape pomace (winery by-product) stabilized by blanching and high hydrostatic pressure has recently been successful at delaying lipid oxidation in burgers. The aim of this study was to investigate whether it can also delay lipid oxidation in dry-cured sausages, and to compare its effect when added at 0.5 and 3% with those of synthetic additives (sodium nitrite and ascorbic acid) and no additives (Control) in lipid and protein oxidation, the instrumental color, the sensory characteristics, and the volatile compounds. The pomace (68.7 ± 7.4 mmol Trolox g^−1^) was as effective as the additives at preventing lipid oxidation, resulting in values 3.2–3.8 times lower than the Control sausages. However, the pomace was not effective at decreasing the microbial counts, improving the instrumental and sensory color and the volatile compound profile, and decreasing the off-odor and off-flavor developed in the Control sausages. The lack of a detrimental effect of the pomace at 0.5% on the volatile compounds and the sensory characteristics and its benefits to delay lipid oxidation suggest that it might be useful to improve the oxidative stability. Conversely, at 3%, with a detrimental effect on some sensory characteristics and no benefits over the lower dosage, is not advisable.

## 1. Introduction

Dry-cured sausages are well-appreciated meat products usually produced by mixing ground meat with seasonings, stuffing, and ripening. It is a common industrial practice to also include some synthetic additives, such as nitrites, to improve microbial stability and safety, color, and flavor [1]. However, new trends in consumers’ preferences are leading to an increased demand for additive-free meat products. In recent years, several attempts have been made to substitute synthetic additives with natural products, such as parsley, celery, chard, and red wine, although, at the moment, no alternatives for nitrites that fulfil simultaneously all of nitrite’s functions (for example in color and flavor development, the oxidative status, and the antimicrobial effect) have been found [2].

Among the natural products that could improve the quality of meat products, as well as enrich them in healthy compounds, such as phenols [3], several by-products are receiving increased attention in the food industry. Grape by-products, produced in large quantities in the winery industry, have been extensively researched [4,5,6]. They have been essayed in food systems as a source of dietary fiber, colorings, and antioxidants [4,5,6,7]. These by-products are rich in phenols, which have a beneficial effect on human health [8] and, when added to food systems, have relevant antioxidant and antimicrobial activities [6,9,10]. These activities might be useful to reduce the use of synthetic additives in meat products, which has been the topic of recent research [11]. 

The addition of both white grape by-products (usually before fermentation) and especially red grape by-products (usually after fermentation) has recently been researched in some meat products. In burgers, red grape pomace provided burgers enriched in phenols but a loss in the overall acceptability [12], and it was proposed as an alternative in burgers to ascorbate to successfully preserve *a** and prevent lipid and protein oxidation [13], to salt to control microbial spoilage [14], to sulfites against radical formation and myosin heavy chain crosslinking but not against thiol loss [15], and to erythorbate to increase the oxidative stability and preserve *a** [16]. In beef sausages, red grape pomace was suitable to partially substitute nitrites in terms of lipid oxidation, without causing a loss in acceptability [11]. In dry-cured sausages, grape-pomace-based extracts have been essayed to substitute nitrites, resulting in a similar prokaryotic community [17] and without differences in lipid oxidation but a loss in color but not in the aroma profile [18], as well as to enrich the sausages in health-beneficial phenols [3] and to replace fat [19]. However, some of these studies have not researched the effect on the microorganisms present in the product.

Grape pomace, the main by-product from the winery industry, is composed of skins, seeds, stems, and the remaining pulp. It contains large quantities of water and microorganisms that reach high counts quickly [20,21], as well as active enzymes, which lead to rapid spoilage. To overcome these issues, grape pomace is usually processed prior to its use in the food industry. This usually involves seed separation and solvent extraction, with the corresponding waste generation, or drying. Drying is the simplest and traditional process for rendering pomace into a safe and stable product. However, it can cause the loss of thermolabile bioactive compounds [22,23]. Contrarily, the application of non-thermal technologies, such as high hydrostatic pressure (HHP), is an alternative to stabilize the by-products with minimal losses [24].

Recently, HHP combined with blanching has been proposed to stabilize red [25] and white grape pomace [20], facilitating its integral use without extraction, drying, or irradiation. HHP, a non-thermal treatment, decreases the microbial counts without causing losses in the phenol content, and thermal blanching inactivates enzymes, such as polyphenoloxidase [20,21]. This stabilized grape pomace successfully delayed lipid and protein oxidation in pork burgers, although it showed a limited antimicrobial or color-protective effect [20]. Lipid oxidation has a deleterious effect in dry-cured meat products, and it is usually the main limitation factor for the shelf-life of the final products, sold either sliced or as whole pieces. The aim of this study was to investigate whether this HPP-treated white grape pomace can also delay lipid oxidation in dry-cured sausages and to compare it with synthetic additives in terms of its effect on the instrumental color, the sensory characteristics, and the volatile compounds.

## 2. Materials and Methods

### 2.1. Stabilized White Grape Pomace Production

Fresh wine pomace from destemmed white grapes (*Cayetana* cv) provided by a local wine manufacturer company located in Alburquerque (Badajoz, Spain) was vacuum-packaged and kept at −80 °C until use. The pomace was stabilized by blanching and applying a high hydrostatic pressure (HHP) treatment previously described [20]. Briefly, the pomace was prepared by blanching with steam at 103 °C to inactivate the polypenoloxidase enzyme, freezing and grinding to a fine powder, vacuum-packaging in plastic bags (polyamid polyethylene 20/100, oxygen permeability of 50 cm^3^ m^−2^, 24 h^−1^, and 0% relative humidity, 120 µm thickness, Eurobag, Málaga, Spain), applying a high hydrostatic pressure treatment at 600 MPa for 5 min to inactivate microorganisms, and freezing at −80 °C until use (approximately one month). The resulting stabilized pomace was characterized in detail [20]. The phenolic compound concentration was 766.7 ± 15.6 mg GAE 100 g^−1^, and the phenolic profile is available [19]. Flavanols were the main bioactive compounds in this white grape pomace, and anthocyanins were absent [20], as it has been reported for white grape pomace [17,18,26].

### 2.2. Manufacture of Salchichón

The sausages were manufactured with pork (60% whole deboned leg pork, 40% backfat) slaughtered on the same day and then refrigerated (approx. 3–4 °C), and 20 g kg^−1^ of fresh garlic, 25 g kg^−1^ of salt, and 2 g kg^−1^ of black pepper. The pork was ground to 8 mm using a meat grinder (PT-82, Mainca, Barcelona, Spain), the garlic using a DJ305110 Moulinex chopper (Moulinex, Ecully, France), and the pepper using a coffee electric grinder. Each batter was mixed using an automatic vacuum mixer (Cato, Girona, Spain).

Four types of salchichón masses (4 kg each) were prepared: Control (plain formulation); NITRASC (including 0.15 g kg^−1^ of sodium nitrite, Panreac, Barcelona, Spain, and 0.5 g kg^−1^ of L-ascorbic acid, Laffort, Guipuzcoa, Spain); 0.5%Pomace (including treated white grape pomace at 0.5%, *w*/*w*); and 3%Pomace (including treated white grape pomace at 3%, *w*/*w*). The pomace percentages were chosen on the basis of previous work in burgers showing that 0.5% was the lower percentage with any beneficial effect, whereas 3% was the higher percentage not causing panelists’ rejection.

After mixing under a vacuum (0.5–0.7 bar, without temperature control, Vacuum Mixer, Talleres Cato SA, Barcelona, Spain) at approx. 120 rpm for 5 min, the batters were kept at 5 °C for approx. 1.5 h before stuffing them into natural dry pork casings of 45–50 mm of diameter previously soaked. The sausages were then kept in an industrial chamber in the pilot plant at our institute at 12–14 °C and at a relative humidity of 85% for 28 days. The total weight losses were 48%, and the final averaged weight for the sausages was 158–221 g. Five sausages per batch were manufactured, with a total of twenty sausages. All the analyses were performed on each sausage.

### 2.3. Chemical and Physical–Chemical Analyses

The total antioxidant activity of the treated white grape pomace was measured after extraction in triplicate by mixing 1 g of pomace with 50 mL of a milliQ water/methanol/citric acid (80:19.9:0.1) solution at 60 °C for 60 min, centrifugation at 10,000 rpm at 4 °C for 15 min, and evaporation for 20 min. MilliQ water was then added to reach a final volume of 50 mL. A total of 50 µL was mixed with 1 mL of ABTS (Sigma-Aldrich, Madrid, Spain). The absorbance was measured immediately at 750 nm in triplicate in a UV-2401 PC Shimadzu spectrophotometer (Shimadzu Scientific Instruments, Columbia, MD, USA). A standard curve with Trolox (Sigma-Aldrich, Madrid, Spain) with concentrations being in the 0–350 µM range was used to express the results as mmol Trolox g^−1^.

The protein and moisture contents were determined according to the AOAC methods [27]. The protein content was determined by the Kjieldahl method. Briefly, sulfuric acid was used for the digestion of samples, and the released nitrogen was determined by titration. Moisture was calculated after drying the samples at 104 °C until constant weight. The fat content was determined gravimetrically, after extraction with chloroform/methanol (2:1) and the subsequent evaporation of solvents [28].

Water activity (*a_w_*) was measured using a Labmaster-*a_w_* meter (Novasina AG, Lachen, Switzerland). The pH value was determined after homogenization with deionized water (1:10) using a Crison pH25+ pH-meter (Crison, Barcelona, Spain). 

Lipid oxidation was measured by using the thiobarbituric acid reactive substance (TBA-RS) method [29] and a standard curve of tetraethoxypropane (Sigma-Aldrich, Madrid, Spain), with concentrations in the 0–0.004 mg mL range, and it was expressed as mg malondialdehyde (MDA) kg^−1^ dry-cured sausage. Protein oxidation was assessed by measuring the carbonyl groups formed during incubation with 2,4-dinitrophenylhydrazine (DNPH) (Panreac, Barcelona, Spain) in 2 N HCl [30], and it was expressed as nmol carbonyls mg^−1^ protein.

The total phenol content was determined using the Folin–Ciocalteu colorimetric method [31] and a standard curve of gallic acid (Merck, Madrid, Spain) with concentrations in the 0–16 ppm range. The results were expressed as mg gallic acid equivalent (GAE) 100 g^−1^ of dry-cured sausage.

### 2.4. Microbial Analyses

Ten grams of each sausage was sampled aseptically and homogenized with 90 mL of peptone water (Merck, 1.07043) in a Stomacher^®^ 400 Circulator laboratory blender. In total, 1 mL of serial decimal dilutions in sterile peptone water was then poured or spread onto total count and selective agar plates and incubated. The *Staphylococcus aureus* counts were determined using Baird Parker Agar (Merck, 1.05406) and incubation at 37 °C for 24–48 h. *Clostridium* spp. was incubated on TSC Agar (Merck, 1.10235) at 37 °C for 24 h. *Escherichia coli* and the total coliforms were incubated on Chromocult Agar (Merck, 1.10426) at 37 °C for 24–48 h. *Salmonella* spp. and *Listeria monocytogenes* were determined according to ISO 6579: 2017 [32] and ISO 11290–1: 2017 [33], respectively. The mesophilic aerobic bacterial counts were determined using a standard Plate Count Agar (Merck, Darmstadt, Germany, 1.07881), and incubation was at 30 °C for 72 h. The lactic acid bacteria were incubated on MRS Agar (Merck 1.10660) at 37 °C for 72 h. The aerobic psychrotrophic bacteria, spread onto Plate Count Agar (Scharlau, Barcelona, Spain), were incubated at 7 °C for 10 days. The mold and yeast counts were determined after spreading on Potato Dextrose Agar and incubation at 25 °C for 5 days.

The results were expressed as log10 CFU (colony forming unit) g^−1^. The detection limit was 10 CFU g^−1^, except for *S. aureus*, whose limit was 100 CFU g^−1^. The absence of *Salmonella* spp. and *L. monocytogenes* in 25 g of sausage was also assessed.

### 2.5. Instrumental Color

The CIELAB instrumental color (*L**, *a**, and *b**) was measured by using a Minolta CM-5 spectrophotometer with autocalibration (Minolta Camera, Osaka, Japan), with a 10° illuminant, a D65 angle, SCI, and a measuring area of 30 mm^3^. Chroma and the Hue angle were calculated as *C** = (*a^⁎^*^2^ + *b^⁎^*^2^)^0.5^ and *h°* = tan (*b**/*a**)^−1^, respectively. The measurements were performed on both sides of one slice (approx. 1 cm thickness) per sausage.

### 2.6. Volatile Compound Analysis

Two grams of ground dry-cured sausage or five grams of grape pomace was weighed into a 20 mL vial, which was then screw-capped with a Teflon-silicone septum. In total, 20 μL of a 4-methyl-1-pentanol methanolic solution (306 μg mL^−1^) was added to each sample (final concentration: 1.22 mg kg^−1^). Each sample was equilibrated for 5 min at 37 °C using a CombiPAL autosampler (CTC Analytics, Zwingen, Switzerland). A 1 cm 50/30 μm DVB/CAR/PDMS SPME fiber (Supelco, Bellefonte, PA, USA) was then inserted into the vial through the septum and exposed to the headspace for 30 min at 37 °C (a temperature sufficient to increase the headspace volatile compound concentration without causing protein denaturation). 

The SPME fiber was inserted into the injection port (set at 270 °C) of a Varian CP-3800 gas chromatograph coupled to a Varian Saturn 2200 MS mass spectrometer (Varian Inc., Palo Alto, CA, USA) equipped with an HP-5 capillary column (30 m × 0.32 mm × 0.25 μm; Agilent Technology, Santa Clara, CA, USA) and with helium (1 mL min^−1^) as the carrier gas. The oven temperature was held at 35 °C for 10 min, raised to 250 °C (7 °C min^−1^), and held for 5 min, with a total running time of 45 min. The temperatures of the transfer line, trap, and manifold were 280 °C, 200 °C, and 60 °C, respectively. The mass spectra were obtained by electronic impact at 70 eV, with one scan s^−1^ over the 40–300 *m*/*z* range. 

The compounds were identified by comparing their mass spectra and linear retention indexes (LRIs) with those of standards (Sigma–Aldrich, St. Louis, MO, USA) or with mass spectra contained in the NIST library by using Agilent MSD Chemstation E.02.01.1177 software and/or LRI reported in the literature [34,35,36]. The concentration was estimated by using the internal standard and expressed as μg kg^−1^.

### 2.7. Sensory Analysis

Twelve panelists trained to take part in descriptive tests (at least approx. 50 h) and with recent experience of this test applied to dry-cured products participated, after two specific training sessions, in a conventional descriptive test to measure, using unstructured 10 cm scales, the perceived intensity of twelve descriptors: lean color (ranging from pale pink to dark red), fat color (from white to dark yellow), “salchichón” odor, off-odor, defective texture, juiciness, saltiness, spicy flavor, flavor intensity and off-flavor (ranging from not perceptible to very intense), and hardness (from very tender to very firm).

All the sessions were carried out in a test room with individual booths at room temperature (22 ± 2 °C) and with white fluorescent lighting. Two slices (approx. 0.5 cm thickness) of each sample were presented to each panelist on a plate. A glass of water (approx. 100 mL of water) at room temperature was also provided. Four sausages (one per group) were randomly assessed in each session, with a total of five sessions. All the panelists had been informed of the experimental setting and consented the use of their answers.

### 2.8. Data Analysis

A one-way (sausage group) analysis of variance (ANOVA) was performed on the data from all the analyses except the sensory test to check for differences between the four groups. When the ANOVA showed a significant effect (*p* ≤ 0.05), pairwise comparisons using Tukey’s test were carried out. For the sensory data, a generalized linear mixed model (fixed effects: sausage group and panelist; random effect: session) was applied. When the effect of the sausage group was significant (*p* ≤ 0.05), the Bonferroni post hoc test was used to compare the groups in pairs. 

A Principal Component Analysis was performed to check the differences between the four groups of sausages, and to investigate the multivariate relationships among variables. The Pearson correlation test was carried out to evaluate the bivariate relationships between variables. The mean value from the panel responses for each sensory trait and each sausage was used in these interdependence analyses. The analyses were performed by using IBM Statistics 27.0.1.0 software (IBM Corp., Chicago, IL, USA).

## 3. Results and Discussion

The results for the Control dry-cured sausages and the sausages with synthetic additives or with white grape pomace stabilized with blanching and high hydrostatic pressure, with antioxidant activity (68.7 ± 7.4 mmol Trolox g^−1^ according to the ABTS test described in Section 2.3), revealed significant differences in the general parameters, the microbial counts, the instrumental color, the volatile compound profile, and the sensory characteristics. 

### 3.1. General Parameters and Phenol Content 

The protein and fat contents were not affected by the addition of synthetic additives or grape pomace (Table 1). The lack of effect of the pomace addition on them could be expected since it was added at low percentages and it had a low content of protein and fat (2.3 and 1.7%, respectively). This is in line with previous results comparing dry-cured sausages with a grape-seed-based extract vs. with sodium nitrite [18]. Conversely, the moisture content and the *a_w_* and pH values were significantly different between the four dry-cured sausage groups (Table 1). 

The values for *a_w_* and the moisture content (Table 1) were within the ranges previously reported in similar products [17,18,26]. Although relatively high, *a_w_* was still below the 0.92 limit for ready-to-eat food established in the European Commission Regulation 2073/2005 [37] to prevent the growth of *Listeria* spp. *a_w_* was significantly lower in the pomace-added sausages than in the NITRASC group, similarly to the trend found for the moisture results (although Tukey’s test was not sufficiently powerful to reveal significant differences between groups). The pomace, despite being rich in water (69.2% [20]), was able to reduce the free and total water content. This effect might be related to the low pH of the pomace (3.96 [20]), which could have caused an immediate drop in pH, similarly to the pH drop reported when it was added at 0.5 and 3% to minced meat [20]. The low pH of the pomace might have favored an earlier protein denaturation and, consequently, a faster water loss at the beginning of the ripening stage. These lower values in *a_w_* indicate that the pomace could accelerate sausage dehydration and shorten slightly the ripening stages that end when a fixed *a_w_* or water loss is reached, although further research is required to confirm this result. There are no previous studies on the use of grape pomace to produce dry-cured sausages. Previous studies adding a grape-seed-based extract to dry-cured sausages reported no effect on the moisture content (*a_w_* not researched) [17,18], which suggests that the different composition of that seed-based extract and our pomace might differently affect water retention during ripening.

The pH values (Table 1) were in the range defined by previous studies for most Mediterranean sausages [17,18,26,38]. As shown in Table 1, the Control and 0.5%Pomace groups reached the highest values, and the NITRASC and 3%Pomace groups the lowest, which reveals that the differences in the final pH cannot be attributed solely to the low pH of the pomace. Usually, dry-cured sausages undergo a drop in pH during the earlier stages of ripening mainly due to the lactic acid bacteria (LAB) activity [39], and a later rise can also occur due to other microorganisms causing proteolysis and consuming the lactic acid previously generated [40,41]. Our results suggest that the differences in pH might be related to a stimulation of LAB and/or an inhibition of proteolytic bacteria caused by the synthetic additives (nitrite and ascorbic acid in the NITRASC group) and the pomace added at 3%, but not at 0.5% (pomace at 3% was not sufficient to cause any difference from the Control group). With respect to the stimulation, it has been suggested that a grape-seed-based extract may favor the drop in pH by promoting the growth of *Lactobacillus* during the first fermentation phase [18]. Regarding the inhibitory effect on proteolytic microorganisms, previous studies have reported antimicrobial activity in several components of grape pomace [6,9] as well as the well-known antimicrobial effect of nitrites [1]. 

The phenol content was significantly different between the four groups, with the highest content in the Control and the 0.5%Pomace groups and the lowest in the NITRASC one (Table 1). The high values found in the Control group (without any grape pomace) suggest that a fraction of the compounds quantified as phenols were generated during sausage ripening. This is in line with our previous results using the same test for phenol quantification, where the phenol content in Control burgers increased from day 1 to day 7 [20]. These compounds generated during sausage ripening and burger storage could include either phenols or other compounds able to interfere with the phenol quantification test. 

Some phenols could have been generated during ripening through microbial activity. Fermentation generally increases the phenolic content [42], and some microorganisms, such as *Brettanomyce* yeasts, are able to transform phenols [43]. In this regard, the groups with the highest phenol content (Control and 0.5%Pomace) had the highest microbial counts, whereas the one with the lowest content (NITRASC) had the lowest one, suggesting that non-reacting compounds (including some phenols) from garlic and black pepper (included in all the groups) might have been transformed into reacting phenols. 

In addition, the generation of compounds interfering with the phenol measurement could also explain the high values in the Control group and the lack of differences between the two pomace groups (with 0.5 vs. 3% pomace). Interferences in the Folin–Ciocalteu method have been previously reported and attributed to reactions involving some synthetic additives, such as sulfites [44,45], but also compounds such as sugars and amino acids [46], the latter being generated through ripening and microbial activity. The occurrence of these interferences could have resulted in an overestimation of the phenolic compound content, leading to weak relationships between it and the activities resulting from these compounds, such as the antioxidant activity. In this regard, the phenolic compound content was not significantly correlated with lipid and protein oxidation (*p* = 0.115 and 0.234, respectively).

It should be noted that the pomace added to the 0.5%Pomace and 3%Pomace groups only provided 3.83 and 22.9 mg phenols 100 g^−1^ of fresh batter, respectively, whereas the final content was 96.8 and 86.8 mg phenols 100 g^−1^ dry-cured sausage, respectively. This reveals that the dry-cured sausages had a high content in compounds quantified as phenols that were not in the pomace. The phenolic compounds of grape pomace are directly related to the antioxidant activity of the pomace, although it is a highly variable parameter [17,18,26]. However, our results suggest that the total phenolic content measured in the dry-cured sausages using the Folin–Ciocalteu reagent might not be a relevant indicator for the pomace added and its phenols and, therefore, for the antioxidant potential in the dry-cured sausages. This suggests that a more specific measurement of the phenols (such as the individual phenol quantification) instead of the total phenol content might be advisable in the dry-cured sausages to have a more precise measurement of the antioxidant compounds from the pomace that remain after the dry-cured sausage ripening stage.

### 3.2. Microbial Counts

With respect to the dry-cured sausage safety, the counts for all the pathogens checked (*S. aureus*, *Cl. perfringens*, *E. coli*) were under the detection levels and, therefore, are not included in Table 2. In addition, the dry-cured sausages presented an absence of *Salmonella* spp. and *L. monocytogenes* in 25 g. 

Regarding the other microorganisms checked, the total coliforms were also under the detection limits and, therefore, not included in Table 2. For the microbial groups above the detection limits, the counts were high (Table 2). Even so, they were still within the ranges reported for similar microbial groups in dry-cured sausages [26,38,39,47]. The high mesophilic and psychrophilic counts could be partly caused by the high counts of LAB in Control and pomace (0.5 and 3%) sausages, since LAB are also detected during mesophilic and psychrophilic analyses. These high counts (Table 2) might have been favored by the relatively high values in pH of the samples (Table 1), as suggested previously [48]. Molds and yeasts were also within the usual range [39].

There were significant differences between the four sausage groups in the mesophilic, psychrotrophic, and LAB counts (Table 2), the NITRASC sausages reaching the lowest values for all of them (Table 2). The differences between the NITRASC and Control groups are in line with previous studies reporting the effectiveness of sodium nitrite against the microorganisms [1,49], and the pomace was not as effective as nitrites to control microbial growth. It should be noted that although low microbial counts are generally advisable, that is not the case for LAB, which are a microbial group involved in dry-cured sausage stabilization and the assurance of its hygienic/sanitary quality [39]. Therefore, the lower LAB counts in the NITRASC group than in the 0.5%Pomace one could not be considered straightforwardly positive since a low LAB growth could potentially favor the development of anomalous fermentation.

No significant differences were found between the Control and the pomace groups, which suggests that the pomace did not have any beneficial effect to achieve lower microbial counts at the end of ripening. These results are in line with a previous study on the same grape pomace applied to burgers, also without differences from the Control burgers in any of those microbial groups, which was attributed to factors such as a decrease in the antimicrobial activity from the pomace due to phenol–protein interactions and the hydrophobicity of phenols in a matrix where the microorganisms are especially present in the more hydrophilic tissues [20]. In this regard, it has been reported that the efficacy of grape pomace against microorganisms, including pathogens, depends on factors such as the pH and polarity of the matrix, the microbial species, the pomace concentration in the product, and the grape cultivar [6,10], and the effect of pomace addition to different products has been reviewed, as well as the mechanism involved in the antimicrobial activity [6,10]. Although a higher dosage might have an effect on the microbial groups, it was ruled out because of its strong detrimental effect on the sensory characteristics, as it was already mentioned. 

Despite the antimicrobial activity of grape pomace [6,10], our results for dry-cured sausages show that they are not effective against the three microbial groups that were over the detection limits, which confirms previous results for our HHP-stabilized white grape pomace on pork burgers [20]. This suggests that this pomace may not be sufficient to control those microorganisms. Further studies are advisable to check whether the stabilized grape pomace is effective at decreasing pathogens, because the analysis performed did not provide enough information on whether the pomace could be useful to control foodborne microorganisms, since pathogens were under the detection limit in all the groups of dry-cured sausage.

### 3.3. Oxidation Status

Lipid oxidation (measured as MDA content) was significantly higher in the Control group than in the others, with values 3.2–3.8 times higher (Table 3). The values for the groups with pomace (0.5%Pomace and 3%Pomace) and sodium nitrite and ascorbic acid (NITRASC) were similar, which reveal that the pomace was as effective as the synthetic additives at hindering lipid oxidation. It should be noted that there were not significant differences between the 0.5 and 3% pomace groups (Table 3) and, therefore, the lower pomace content was sufficient to reach the maximum effect. This indicates that the antioxidant activity of the grape pomace (68.7 ± 7.4 mmol Trolox g^−1^) was effective at preventing oxidation when added at a low dosage to a food matrix where interactions of phenols with proteins might happen, as well as changes during the ripening stage.

The decrease in lipid oxidation caused by the pomace is in line with a previous study applying the same pomace to burgers, which reported less lipid oxidation in the samples with 0.5, 1, or 3% pomace than in the Control samples and even than in burgers with sulfites [20]. They also match previous results on a grape-seed-based extract (with added hydroxytyrosol and tocopherol), which resulted in similar values to sodium nitrite in the lipid oxidation of dry-cured sausages [18]. The decrease in lipid oxidation might be relevant to increase the shelf-life of the sausages. This is especially advisable when additional oxidative factors are involved, such as slicing, which is increasingly performed to meet new consumers’ preferences for packaged sliced dry-cured products.

The antioxidant activity of grape pomace has already been demonstrated [4,5,6]. This effect was proved as useful in food systems, including muscle foods such as burgers and salmon [13,50]. Our study demonstrates that white grape pomace stabilized by blanching and HHP can be used to improve the lipid stability of dry-cured sausages, and it can be a green alternative to synthetic antioxidants.

As for protein oxidation (measured as carbonyl content), no differences between the four sausage groups were found (Table 3), the pomace and the synthetic additives not being useful to prevent protein oxidation. Despite lipid and protein oxidation being generally intertwined, our results showed that there was not a strong correlationship (R: 0.394, *p* = 0.086). Previous studies have also reported that both parameters are not always affected to the same extent by several factors in dry-cured products [48,51], since they measure compounds generated at different stages of the oxidation reactions. The lack of effect of the pomace on protein oxidation (Table 3) matches the results from a previous study applying the same pomace to burgers, which reported no differences after 7-day storage, and an inconsistent effect before storage [20]. Previous studies including grape-based products in dry-cured meat products have not researched their effect on protein oxidation [17,18,19]. For other meat products, the results have been inconsistent: for example, grape by-product extracts lowered the free thiol content in beef patties [50] but had no effect on lamb patties [13] even though in both studies they delayed lipid oxidation. Further studies focused on protein oxidation are advisable to broaden the knowledge on the oxidation phenomena occurring in dry-cured products when phenols or grape pomace are added. To sum up, the results suggest that despite the prevention of lipid oxidation (Table 3) and the activity of some phenols to prevent protein oxidation [52], the pomace had no noticeable effect to hinder protein oxidation in dry-cured sausages.

### 3.4. Instrumental Color

The instrumental color was significantly different between the groups, specifically in *a**, *b**, and *h°*. The NITRASC group reached the highest values in *a** and the lowest in *b** and *h°*. The lowest values in *a** in the nitrite-free sausages reveal that the pomace addition cannot compensate for the absence of nitrites. In addition, the pomace-added groups were not different from the Control group in any instrumental color coordinate (Table 4), which suggests that the pomace does not have any beneficial effect (either at 0.5 or at 3%) to improve dry-cured sausage color.

The lack of effect of the pomace is partially in line with those reported for pork burgers with the same pomace after 7-day storage, with no effect on *L** and *a** but higher values in *b** in the samples with pomace at 3% than in the Control ones (on day 1, the effect was only significant on *h°*) [20].

The results are roughly in line with previous studies comparing *L**, *a**, and *b** [18] and *C** and *h°* [17] of dry-cured sausages with sodium nitrite vs. with a grape-seed-based extract. The lack of differences in *L** between the pomace and NITRASC groups (Table 4) is in line with those previous results [18]. However, in contrast to our results, similar values for *a**-, *b**- [18], and *a**-based color parameters *C** and *h°* [18] between sausages manufactured with a grape-seed-based extract and with nitrites were found. Those similarities implied that the extract and nitrites influenced color development to a similar extent, even though grape by-products are not rich in nitrites. It was suggested that color might have developed in the extract-containing sausages through an alternative pathway involving zinc (yielding the stable red compound Zn-protoporphyrin) instead of nitrites (which result in nitrosomyoglobin) [18], which has already been described in Parma ham [53,54]. However, our results indicate that those reactions were not noticeable when white grape pomace was used.

The lack of agreement between the results in Table 4 and those studies might be due to the differences between the nitrite-containing sausages (theirs included only sodium nitrite, whereas ours also included L-ascorbic acid) and the compounds and grape products included in the related sausages (they added a grape seed extract, hydroxytyrosol, and tocopherol [17,18]), whereas we added a stabilized grape pomace from pressed grapes after wine production including seeds, skin, and pulp). In addition to differences in the content in grape compounds promoting Zn-protoporphyrin formation, further differences might be involved. In this regard, differences in the microorganisms might have been relevant. The microorganisms are involved in color development in dry-cured products through different mechanisms: they are involved in changes in pH (which affect the color reactions), in nitrate and nitrite reduction (which favors color reaction development), in the peroxide generation (peroxide accumulation leads to iron oxidation and discoloration), and in the catalase activity (which degrades the peroxides), with microorganisms such as staphylococci increasing this catalase activity [39]. In this respect, our results showed that the instrumental color and the microbial counts were related. *a** and *h°* were correlated with all the microbial groups included in Table 2: the mesophilic and psychrophilic bacteria, the LAB, and molds and yeast were correlated with *a** (−0.619, −0.635, −0.449, and 0.452, respectively, *p* = 0.003−0.045) and *h°* (0.593, 0.650, 0.480, −0.474, respectively, *p* = 0.002−0.035); conversely, *L**, *b**, and *C** were not correlated with any microbial group. These correlationships might reveal either a direct effect of those microorganisms on *a** or a similar trend resulting from a common source of variation, namely a common factor causing low *a** and also favoring high counts in most microbial groups.

To sum up, the grape pomace was not effective at improving the instrumental color of dry-cured sausages and, therefore, it is not an alternative to nitrites in terms of color. This indicates that a strategy aimed at producing nitrite-free sausages with the HHP-stabilized pomace should also include additional actions to improve color.

### 3.5. Volatile Compounds

Thirty-four volatile compounds were tentatively identified in the dry-cured sausages (Table 5). Six of them were also identified in the grape pomace headspace (ethanol, phenylmethanol, 3-methylbutan-1-ol, hexanal, 2-phenylacetaldehyde, and nonanal) (Table 6), whereas twenty-one compounds of the pomace (including the two most abundant compounds) were not found in the sausages. The lack of some pomace compounds in the sausages might be explained in terms of the small percentage of pomace added (which caused a 33.3–200-time dilution), and the loss of the volatile compounds caused by their release to the atmosphere and their involvement in chemical reactions (including the oxidation of alcohols) during the ripening stage of the sausage production.

Seven of the thirty-four volatile compounds were significantly different between the four groups (Table 5): 3-hydroxybutan-2-one, 2,3-butanediol, hexanal, α-fellandrene, 2-phenylacetaldehyde, nonanal, and α-caryophyllene. The pomace at 0.5% did not cause differences from the Control group in any compounds, whereas at 3%, differences appeared in 2,3-butanediol and hexanal. The differences between the pomace groups and the NITRASC one were larger: four compounds differed when it was added at 0.5% (3-hydroxybutan-2-one, hexanal, 2-phenylacetaldehyde, and α-caryophyllene), and three when it was added at 3% (2,3-butanediol, hexanal, and 2-phenylacetaldehyde). 

Only three of the seven compounds different between the groups were identified in the pomace, whereas the others were not. Regarding those three compounds, all of them were aldehydes that can be generated through lipid oxidation reactions, and they followed a different trend: hexanal and 2-phenylacetaldehyde reached the lowest content in the NITRASC group, whereas nonanal reached the highest. 

Hexanal, an estimator for lipid oxidation, had the highest estimated content in the 3%Pomace group, with a sharp increase (almost tripled) with respect to the Control one, whereas the lowest content appeared in the NITRASC group. Two phenomena might be mainly involved in these differences: lipid oxidation reactions in the sausages, and the hexanal content in the added pomace. Regarding lipid oxidation, the results for hexanal were not correlated with the results from the TBARS test (*p* = 0.248). This might be due to the fact that the TBARS test did not measure hexanal but malondialdehyde instead, which is generated at a different stage of the lipid oxidation reactions. Regarding the hexanal content of the pomace (281 µg/kg^−1^ pomace), it was not sufficient to triple the sausage content (650.0 vs. 225.5 µg kg^−1^ sausage in the 3%Pomace and Control groups, respectively) or double the values when added at 3% instead of at 0.5% (650.0 vs. 298.8 µg kg^−1^ sausage, respectively). However, a relevant contribution of the pomace should not be ruled out since it contains not only hexanal but also its precursors. Hexanal has been identified as an odorant of dry-cured meat products [18,55], and, therefore, changes in it are expected to be critical for the sausage odor and flavor. However, it was not correlated with the off-odor and off-flavor (*p* = 0.699 and 0.233), although it was with juiciness and defective texture (R: −0.663 and 0.919, *p* = 0.001 and <0.001, respectively).

2-Phenylacetaldehyde, generated from amino acid degradation reactions as well as lipid oxidation, followed a similar trend to hexanal, except for the lack of differences between the two pomace groups. This suggests, again, that the pomace might not have been the main source of this compound, and that its production might have occurred in the sausages during ripening. This compound has been identified as a dry-cured meat product odorant [56] and, therefore, changes in it might influence the sausage odor and flavor, which is supported by the significant correlationships with the off-odor, flavor intensity, and off-flavor (R: 0.463, −0.475, and 0.530, *p* = 0.004, 0.034, and 0.016, respectively). The results for hexanal and 2-phenylacetaldehyde are in line with a previous study also reporting higher values in dry-cured sausages with a grape-seed-based extract than with sodium nitrite [18]. This suggests that the grape-based products, regardless of their form (either HHP-stabilized pomace or a grape seed extract), might not provide the conditions for the sausages to develop the flavor reactions (involving, for example, lipid oxidation and fermentation) exactly as they occur in the nitrite-containing dry-cured products.

Nonanal showed a different trend, with the lowest values in the Control group, the highest in the NITRASC one, and intermediate values in the pomace groups. The different trend between these aldehydes might be related to differences in the pathways leading to aldehyde generation. This compound was not identified in the only previous study that investigated the volatile compounds of dry-cured sausages with a grape-seed-based extract [18]. This compound has not been usually identified as a dry-cured meat product odorant [18,55,56], and, therefore, changes in it are expected to not be critical for the sausage odor and flavor. The limited direct impact of the volatile compounds of the pomace on the volatile compounds of the dry-cured sausage is in line with the sensory results, which did not show any difference between the Control group and the groups with pomace in the odor and flavor traits (Table 6).

With respect to the four compounds different between the groups (Table 5) and not identified in the pomace (Table 6), two of them (3-hydroxybutan-2-one and 2,3-butanediol) are generated through carbohydrate metabolism, whereas the other two (α-phellandrene, α-caryophyllene) are species constituents. 3-Hydroxybutan-2-one and 2,3-butanediol followed a similar trend to 2-phenylacetaldehyde and hexanal, respectively, both with the lowest values in the NITRASC group. They are likely to have been generated through microbial activity, whose effective inhibition caused by sodium nitrite might explain the low values in the NITRASC group. 2,3-Butanediol was especially abundant in the 3%Pomace group, which suggests that a high pomace content might favor the development of fermentation reactions. This suggests that it might be an indicator of detrimental reactions involving microorganisms. 2,3-Butanediol was closely related to the sensory characteristics, especially with the defective texture (0.919, *p* < 0.001), which suggests that it was related to undesired changes in the product. The results for 3-hydroxybutan-2-one and 2,3-butanediol are partially in line with a previous study reporting higher values for the latter in dry-cured sausages with sodium nitrite than with a grape seed extract but no differences for the former [18]. Both volatile compounds have not been usually identified as odorants in dry-cured products [18,55,56], which suggests a weak influence on the sensory traits. That was the case of 3-hydroxybutan-2-one (not correlated with any sensory trait), whereas 2,3-butanediol was correlated with juiciness and the defective texture (R: −0.487 and 0.917, *p* = 0.029 and <0.001) but not with the odor and flavor traits.

α-Caryophyllene had the highest values in the NITRASC group, similarly to nonanal, and the 3%Pomace group. The results suggest that some reactions during ripening, probably involving microorganisms and/or oxidation, might have depleted the initial amount of this compound in the groups with the highest pH (both variables were correlated—R: −0.699, *p* = 0.001). The results are not in line with the lower values reported in dry-cured sausages with sodium nitrite than with a grape-seed-based extract [18], which was attributed to an irregular distribution of the pepper powder and differences in the storage time [17]. α-Phellandrene was significantly different between the four groups according to the ANOVA results. However, Tukey’s test was not powerful enough to show significant differences in the pairwise comparison (Table 5). The lack of differences in α-phellandrene between the pomace and NITRASC groups is in line with a previous study using a grape-seed-based extract [18]. Neither of them had been usually identified as an odorant of dry-cured meat products [18], and neither of them was correlated with any odor or flavor sensory characteristics.

The limited effect of the pomace on the volatile compound profile is roughly in line with previous results on dry-cured sausages with a grape-seed-based extract [18]. The lack of differences between the Control and the 0.5%Pomace group suggests that the pomace at a low dosage might have neither a beneficial nor a detrimental effect on the olfactory traits (odor and flavor). 

### 3.6. Sensory Characteristics

Six out of the eleven sensory characteristics (lean and fat color, off-odor, juiciness, defective texture, and off-flavor) were significantly different between the four sausage groups (Table 7). In addition, the panelist effect was significant for all the variables, whereas the session effect was not for any variable.

The NITRASC group reached the highest values for the lean color and juiciness, and the lowest for the fat color, the off-odor intensity, the defective texture, and the off-flavor. High values in both the lean color (a darker red color) and juiciness are desirable, as well as low values in the fat color (high values correspond to dark yellow, which is generally related to discoloration and lipid oxidation) and the defect (off-odor and flavor and defective texture). This indicates that the synthetic additives provide noticeable benefits from a sensory point of view with respect to the grape pomace or not adding any of them.

The Control group was not significantly different from the 0.5%Pomace group in any sensory traits, and only differed from the 3%Pomace group in the defective texture, where the pomace had a detrimental effect. The results from a Principal Component Analysis (PCA) on the chemical and sensory data show a similar trend, the NITRASC sausages appearing as clearly separated from the others, which were plotted in overlapped areas (Figure 1a).

The results for the lean color match those found for *a** (Table 4), both with the highest values in the NITRASC group and no differences between the Control and the pomace-added groups. In fact, both variables had large negative scores in the first principal component (PC 1) (Figure 1b), and were correlated (R: 0.606, *p* = 0.005). Similarly, the results for the fat color matched those for *b**, with both variables with large scores in the PC 1 (Figure 1b) and correlated (R: 0.491, *p* = 0.028). As mentioned before for *a**, this high value for the lean color in the NITRASC group and the lack of differences between the Control group and the groups with pomace indicate that the pomace is not an effective alternative to nitrites to improve color.

Regarding the off-odor and off-flavor, they were strongly related, appearing in Figure 1b with large scores in PC 1 and correlated (R: 0.804, *p* < 0.001). The Control group reached the highest values and the NITRASC the lowest, whereas the pomace groups had intermediate values statistically similar to them (Table 7). The effect of the grape pomace was not statistically different from the effect of the synthetic antioxidants, but it was not sufficient to decrease the off-odor developed in the Control samples (Table 7). The results followed a roughly similar trend to pH and *b** (Table 3). In fact, Figure 1b shows that both traits were closely related to pH, *b**, and protein oxidation. In fact, there were significant correlationships between the off-odor and pH (0.545, *p* = 0.013), and between off-odor and off-flavor, and *b** (0.533 and 0.506, *p* = 0.016 and 0.023, respectively) and also between off-odor and off-flavor, and protein oxidation (0.603 and 0.591, *p* = 0.005 and 0.006, respectively). Both characteristics were scarcely correlated with the volatile compounds; they were only correlated with 2-phenylacetaldehyde (R: 0.463 and 0.530, *p* = 0.040 and 0.016, respectively), and the off-odor also correlated with disulfide, methyl 2-propenyl (R: −0.499, *p* = 0.025). This suggests that the off-odor and off-flavor (both with low values) were not caused by any compounds from the grape pomace. 

Juiciness reached the highest values in the NITRASC group and the lowest in the 3%Pomace group, which also had the highest value for the defective texture (Table 7). Conversely, when the pomace was added at 0.5%, no detrimental effect appeared in the texture traits. This reveals that the pomace added at 3% may be an issue for consumers since both traits are key parameters for the quality of meat products, whereas at 0.5%, it might not be an issue. A previous study reported an abnormal appearance and flavor in dry-cured sausages produced with grape-seed-based commercial extracts at 0.5%, either a dry one or a leucocyanidin one [3], which led to ruling out both extracts. In our study, the grape pomace at 0.5% did not show such an effect, probably because it was not as concentrated as those extracts.

To sum up, the pomace was not as effective as the synthetic additives at improving lean and fat color and at preventing off-odor and off-flavor development, and the only difference caused by the pomace addition with respect to the Control samples was an increase in the defective texture when added at 3%, without sensory differences when added at 0.5%. The lack of a detrimental effect of the lower pomace dosage on the sensory characteristics (and therefore on what consumers are expected to perceive) and its benefits to prevent lipid oxidation suggest that it could be a feasible option to improve the sausage quality by adding a grape by-product processed without chemicals. Conversely, the higher pomace dosage, with a detrimental effect on some key sensory characteristics and a lack of benefits, would not be advisable.

## 4. Conclusions

The antioxidant activity of the stabilized white grape pomace was sufficient to prevent lipid oxidation, being as effective as the synthetic additives and a green alternative. This could be relevant to increase the shelf-life of the sausages, which is especially advisable when additional oxidative factors are involved, such as slicing before packaging, which is increasingly demanded by consumers.

However, regardless of the dosage, the pomace was not an alternative to nitrites to decrease the microbial counts, to improve the instrumental and sensory color and the volatile compound profile, and to decrease the off-odor and off-flavor. The lack of a detrimental effect of the lower pomace dosage on the sensory characteristics (and therefore on what consumers are expected to perceive) and its benefits to prevent lipid oxidation suggest that it could be a feasible and green option to improve the sausage quality. Conversely, the higher pomace dosage, with a detrimental effect on some key sensory characteristics and a lack of benefits over the lower dosage, is not advisable. Further work would be advisable to investigate whether this blanched and HHP-stabilized pomace might be effective to increase food safety and/or partially replace nitrites. 

## Figures and Tables

**Figure 1 foods-13-00687-f001:**
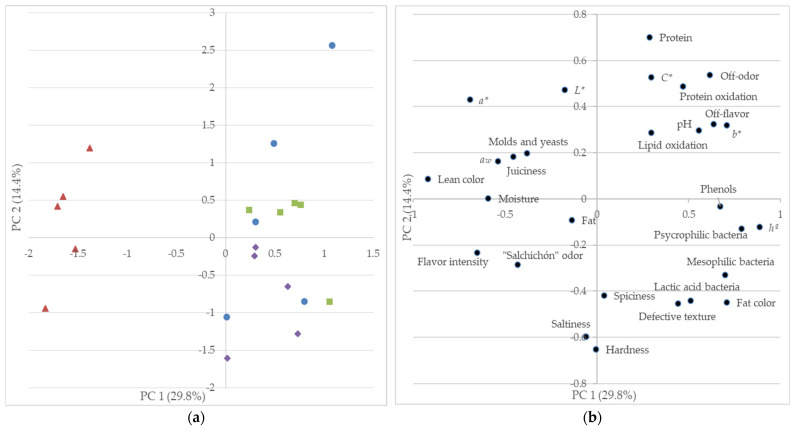
Projection of the samples (**a**) and variables (**b**) onto the space defined by the first two principal components (PC1/PC2) extracted from the variables in Table 1, Table 2, Table 3, Table 4 and Table 5. 
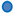
: Control; 
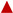
: NITRASC; 
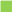
: 0.5%Pomace; 
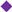
: 3%Pomace.

**Table 1 foods-13-00687-t001:** General parameters and phenolic content of Control dry-cured sausages and dry-cured sausages with synthetic additives (NITRASC) or with stabilized grape pomace at 0.5% (0.5%Pomace) or 3% (3%Pomace).

	Control	NITRASC	0.5%Pomace	3%Pomace	*p*
Protein (%)	33.3 ± 1.4	32.8 ± 1.4	33.8 ± 0.6	32.9 ± 0.7	0.460
Fat (%)	20.9 ± 3.4	21.1 ± 0.8	20.9 ± 1.9	19.8 ± 1.9	0.786
Moisture (%)	29.4 ±1.3	31.0 ± 0.7	28.9 ± 1.0	28.9 ± 1.6	0.039
*a_w_*	0.88 ± 0.0 ab	0.88 ± 0.0 a	0.86 ± 0.0 b	0.86 ± 0.0 b	0.026
pH	5.9 ± 0.0 a	5.6 ± 0.0 b	5.9 ± 0.0 a	5.5 ± 0.1 b	<0.001
Phenolic compounds (mg 100 g^−1^)	93.8 ± 10.2 a	76.9 ± 5.3 b	96.8 ± 9.8 a	86.8 ± 6.7 ab	0.007

The results are expressed as mean ± standard deviation and significance (*p*) from a one-way ANOVA. Different letters in the same row indicate significant differences in Tukey’s test.

**Table 2 foods-13-00687-t002:** Microbial counts (log CFU g^−1^) over the detection limit in Control dry-cured sausages and dry-cured sausages with synthetic additives (NITRASC) or with stabilized grape pomace at 0.5% (0.5%Pomace) or 3% (3%Pomace).

	Control	NITRASC	0.5%Pomace	3%Pomace	*p*
Mesophilic	8.7 ± 0.1 a	8.2 ± 0.2 b	8.7 ± 0.3 a	8.6 ± 0.1 a	0.002
Psychrophilic	8.9 ± 0.1 a	8.2 ± 0.4 b	9.2 ± 0.4 a	8.8 ± 0.1 a	<0.001
Lactic acid bacteria	8.5 ± 0.1 ab	8.5 ± 0.1 b	8.8 ± 0.3 a	8.7 ± 0.1 ab	0.024
Molds and yeasts	5.3 ± 0.5	5.7 ± 0.3	5.4 ± 0.2	5.0 ± 0.6	0.127

The results are expressed as mean ± standard deviation and significance (*p*) from a one-way ANOVA. Different letters in the same row indicate significant differences in Tukey’s test.

**Table 3 foods-13-00687-t003:** Lipid and protein oxidation of Control dry-cured sausages and dry-cured sausages with synthetic additives (NITRASC) or with stabilized grape pomace at 0.5% (0.5%Pomace) or 3% (3%Pomace).

	Control	NITRASC	0.5%Pomace	3%Pomace	*p*
Lipid oxidation (mg MDA kg^−1^)	1.9 ± 0.4 a	0.6 ± 0.0 b	0.6 ± 0.1 b	0.5 ± 0.1 b	<0.001
Protein oxidation (nmols carbonyls mg protein^−1^)	3.0 ± 0.6	2.5 ± 0.3	2.8 ± 0.2	2.8 ± 0.4	0.345

The results are expressed as mean ± standard deviation and significance (*p*) from a one-way ANOVA. Different letters in the same row indicate significant differences in Tukey’s test.

**Table 4 foods-13-00687-t004:** Instrumental color of Control dry-cured sausages and dry-cured sausages with synthetic additives (NITRASC) or with stabilized grape pomace at 0.5% (0.5%Pomace) or 3% (3%Pomace).

	Control	NITRASC	0.5%Pomace	3%Pomace	*p*
*L**	47.3 ± 1.7	48.0 ± 1.8	47.3 ± 2.1	46.6 ± 2.0	0.738
*a**	4.8 ± 1.3 b	6.9 ± 0.7 a	4.9 ± 0.5 b	4.6 ± 1.1 b	0.005
*b**	8.8 ± 1.3 a	6.2 ± 1.5 b	9.3 ± 1.6 a	8.3 ± 1.4 ab	0.018
*Chroma*	10.1 ± 1.7	9.3 ± 1.4	10.5 ± 1.3	9.5 ± 1.7	0.608
*Hue*	62.1 ± 4.9 a	41.2 ± 5.6 b	61.9 ± 5.4 a	61.4 ± 4.8 a	<0.001

The results are expressed as mean ± standard deviation and significance (*p*) from a one-way ANOVA. Different letters in the same row indicate significant differences in Tukey’s test.

**Table 5 foods-13-00687-t005:** Volatile compounds (µg kg^−1^) in Control dry-cured sausages and dry-cured sausages with synthetic additives (NITRASC) or with stabilized grape pomace at 0.5% (0.5%Pomace) or 3% (3%Pomace).

	Control	NITRASC	0.5%Pomace	3%Pomace	*p*
*Alcohols*					
Ethanol	47.1 ± 29.8	47.9 ± 35.2	39.7 ± 26.2	71.2 ± 22.8	0.371
2-Methylthiirene	536.4 ± 269.5	586.4 ± 289.7	655.7 ± 122	928.1 ± 276.5	0.100
3-Methylbutan-1-ol	31.8 ± 19.2	75.6 ± 113.1	35.2 ± 19.5	39.0 ± 26.6	0.636
Butane-2,3-diol	430.3 ± 153.4 b	111.0 ± 88.9 b	468.7 ± 49.2 b	1048.1 ± 358.2 a	<0.001
1-Octen-3-ol	8.7 ± 2.5	7.6 ± 16.9	13.3 ± 16.3	9.2 ± 3.1	0.883
Phenylmethanol	37.0 ± 21.6	32.0 ± 29.3	11.3 ± 15.9	29.1 ± 9.9	0.255
*Aldehydes*					
Hexanal	225.5 ± 138.6 bc	75.8 ± 44.0 c	298.8 ± 100.7 b	650.0 ± 101.3 a	<0.001
2-Phenylacetaldehyde	30.4 ± 6.3 a	2.2 ± 4.9 b	37.4 ± 12.1 a	25.4 ± 8.2 a	<0.001
Nonanal	26.5 ± 10.1 b	56.0 ± 21.6 a	44.3 ± 4.0 ab	46.9 ± 11.9 ab	0.022
*Sulfur-containing compounds*					
Sulfide, allyl methyl	275.1 ± 39.6	203.5 ± 108.1	236.5 ± 65.2	276.4 ± 48.5	0.333
Diallyl sulfide	433.2 ± 110.1	290.6 ± 64.5	345.2 ± 158.1	352.1 ± 159.4	0.403
Disulfide, methyl 2-propenyl	290.4 ± 34.1	385.3 ± 106.6	302.3 ± 82.4	401.4 ± 36.3	0.056
Diallyl disulfide	1069.5 ± 187.8	1060.7 ± 292.0	1210.6 ± 272.8	1429.2 ± 161.9	0.082
Diallyl trisulfide	6.8 ± 1.8	7.7 ± 3.4	9.1 ± 6.9	9.7 ± 6.0	0.790
*Terpenes*					
δ-Phellandrene	38.8 ± 18.3	29.9 ± 4.6	23.0 ± 5.1	29.1 ± 9.4	0.187
α-Pinene	544.6 ± 90.6	588.4 ± 300.4	565.6 ± 90.9	676.5 ± 138.5	0.666
Camphene	33.0 ± 6.4	42.0 ± 12.1	32.8 ± 4.7	39.7 ± 8.5	0.238
β-Pinene	717.6 ± 120.4	913.1 ± 227.8	738.6 ± 92.6	883.8 ± 171.2	0.171
Myrcene	222.6 ± 118.2	326.3 ± 77.9	230.4 ± 133.2	336.7 ± 54.4	0.185
α-Fellandrene	334.5 ± 70.2	298.8 ± 79.8	351.9 ± 61.3	439.4 ± 72.3	0.039
3-Carene	2069.4 ± 354.2	2593.2 ± 636.6	2119.5 ± 178.1	2571.2 ± 406.7	0.129
α-Terpinene	41.1 ± 7.8	36.8 ± 10.8	36.0 ± 6.6	47.6 ± 7.6	0.152
Limonene	1408.2 ± 260.8	1704.6 ± 398.2	1430.7 ± 177.1	1707.7 ± 224.5	0.191
β-Terpinene	3.5 ± 3.9	3.0 ± 1.9	3.1 ± 1.7	3.4 ± 2.0	0.989
γ-Terpinene	15.5 ± 11.4	17.4 ± 4.0	14.6 ± 2.1	17.6 ± 2.8	0.853
α-Terpinolene	25.4 ± 5.3	31.8 ± 7.9	27.9 ± 3.3	32.7 ± 3.5	0.145
β-Terpinolene	55.6 ± 12.3	73.0 ± 20.2	59.2 ± 8.0	71.7 ± 8.7	0.127
δ-Elemene	15.3 ± 3.9	19.8 ± 4.2	17.0 ± 4.0	20.4 ± 1.5	0.129
α-Copaene	21.9 ± 4.7	27.7 ± 5.5	23.5 ± 4.0	28.9 ± 2.4	0.064
β-Caryophyllene	138.3 ± 29.9	178.6 ± 34.4	153.1 ± 28.5	180.1 ± 16.8	0.087
α-Caryophyllene	6.1 ± 1.4 ab	8.0 ± 1.5 a	5.7 ± 1.3 b	8.2 ± 0.8 a	0.010
*Others*					
Pentane	126.4 ± 54.7	147.2 ± 89.2	145.0 ± 34.6	134.2 ± 75.5	0.956
3-Hydroxybutan-2-one	240.3 ± 68.5 ab	101.6 ± 102.5 b	386.0 ± 149.3 a	218.5 ± 76.2 ab	0.005
o-Cymene	279.3 ± 55.7	284.8 ± 153.0	276.8 ± 35.8	345.3 ± 49.1	0.562

The results are expressed as mean ± standard deviation and significance (*p*) from a one-way ANOVA. Different letters in the same row indicate significant differences in Tukey’s test.

**Table 6 foods-13-00687-t006:** Volatile compounds (μg kg^−1^) isolated in the stabilized white wine pomace.

*Alcohols*		*Esters*	
Ethanol	71.3 ± 61.9	Ethyl hexanoate	12.1 ± 2.1
3-Methylbutan-1-ol	5.0 ± 8.6	Ethyl octanoate	15.2 ± 3.0
(Z)-3-Hexen-1-ol	113.2 ± 21.8	Ethyl decanoate	2.3 ± 4.0
(E)-2-Hexen-1-ol	734.9 ± 96.9	Hexanedioic acid, dibutyl ester	8.8 ± 8.4
Hexan-1-ol	862.2 ± 114.5	*Terpenes*	
2-Ethyl-2-hexen-1-ol	46.8 ± 7.1	Alcanfor	13.9 ± 24.0
Phenylmethanol	4.5 ± 4.2	*Others*	
2-Phenylethanol	20.2 ± 0.9	Acetic acid	45.2 ± 44.5
*Aldehydes*		Oxime-methoxy-phenyl	448.4 ± 122.8
Hexanal	281.3 ± 55.6	2,2,4,6,6-Penta-methyl-heptane	28.7 ± 3.3
2-Phenylacetaldehyde	156.6 ± 23.7		
Benzaldehyde	76.0 ± 10.7		
Nonanal	48.7 ± 6.4		
(E)-Hex-2-enal	97.0 ± 11.9		

The results are expressed as mean ± standard deviation from three replicates.

**Table 7 foods-13-00687-t007:** Sensory characteristics of Control dry-cured sausages and dry-cured sausages with synthetic additives (NITRASC) or with stabilized grape pomace at 0.5% (0.5%Pomace) or 3% (3%Pomace).

	Control	NITRASC	0.5%Pomace	3%Pomace	*p*
Lean color	4.6 ± 0.4 b	6.9 ± 0.3 a	4.6 ± 0.5 b	4.7 ± 0.8 b	<0.001
Fat color	3.8 ± 0.5 a	2.6 ± 0.5 b	3.6 ± 0.5 a	3.9 ± 0.6 a	<0.001
“Salchichón” odor	6.3 ± 0.3	6.4 ± 0.2	6.2 ± 0.2	6.3 ± 0.3	0.960
Off-odor	0.5 ± 0.4 a	0.1 ± 0.0 b	0.4 ± 0.2 ab	0.2 ± 0.1 ab	0.002
Hardness	5.2 ± 0.8	5.2 ± 0.4	5.0 ± 0.6	5.6 ± 0.5	0.093
Juiciness	5.0 ± 0.5 ab	5.2 ± 0.3 a	5.0 ± 0.6 ab	4.5 ± 0.3 b	0.050
Defective texture	0.4 ± 0.3 b	0.1 ± 0.0 b	0.4 ± 0.2 b	1.7 ± 0.4 a	<0.001
Saltiness	6.4 ± 0.7	6.3 ± 0.5	6.4 ± 0.4	6.4 ± 0.4	0.394
Spiciness	6.2 ± 0.6	6.1 ± 0.4	6.2 ± 0.6	6.0 ± 0.6	0.437
Flavor intensity	6.3 ± 0.3	6.7 ± 0.3	6.3 ± 0.2	6.2 ± 0.2	0.368
Off-flavor	0.5 ± 0.5 a	0.1 ± 0.1 b	0.4 ± 0.2 ab	0.3 ± 0.2 ab	0.012

The results are expressed as mean ± standard deviation and significance (*p*) from a generalized linear mixed model. Different letters in the same row indicate significant differences in the Bonferroni test.

## Data Availability

The original contributions presented in the study are included in the article, further inquiries can be directed to the corresponding author.

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
