# Peer review of "High-Hydrostatic-Pressure-Stabilized White Grape Pomace to Improve the Oxidative Stability of Dry-Cured Sausages (“Salchichón”)"

_foods, 2024, doi:10.3390/foods13050687_

Round 1

Reviewer 1 Report

Comments and Suggestions for Authors

The aim of the study was to compare the effect of different amounts of HHP-Stabilized White Grape Pomace as additive on dry-cured Sausages. The objectives and background were well stated. However, there are a few comments that need to be addressed prior to the publication.

1. The number of keywords is too many. It is recommended to delete unimportant keywords.

2. In the abstract, it is recommended to add "how many times lower of lipid oxidation was observed in sausages with pomace than that in the control group". Refer to line 324.

3. Line 79, why is the amount of white grape pomace added 0.5% and 3%?

4. Line190, how is the value of 68.7 ± 7.4 mmol Trolox g−1 obtained? Please explain further.

Author Response

The aim of the study was to compare the effect of different amounts of HHP-Stabilized White Grape Pomace as additive on dry-cured Sausages. The objectives and background were well stated. However, there are a few comments that need to be addressed prior to the publication.

We appreciate the kind comments and suggestions to improve the document.

  1. The number of keywords is too many. It is recommended to delete unimportant keywords.

According to the “Instructions for authors” in the Foods website, “Three to ten pertinent keywords need to be added…”. We included eight, which is within the proposed range, and we think they might all be useful for information retrieval. However, some of them could be easily removed if required.

  1. In the abstract, it is recommended to add "how many times lower of lipid oxidation was observed in sausages with pomace than that in the control group". Refer to line 324.

We thank the suggestion. We have included it in the abstract.

  1. Line 79, why is the amount of white grape pomace added 0.5% and 3%?

The following text has been added: “The pomace percentages were chosen on the basis of previous work in burgers showing that 0.5% was the lower percentage with any beneficial effect, whereas 3% was the higher percentage not causing panelists’ rejection”.

  1. Line190, how is the value of 68.7 ± 7.4 mmol Trolox g−1 obtained? Please explain further.

To make this point clear, “(68.7 ± 7.4 mmol Trolox g−1)” has been replaced with “(68.7 ± 7.4 mmol Trolox g−1 according to the ABTS test described in section 2.3.)”

Reviewer 2 Report

Comments and Suggestions for Authors

Author Response

This paper entitled “HHP-Stabilized White Grape Pomace as an Alternative for Additives in Dry-Cured Sausages (“Salchichón”)”.The authors provide intriguing and valuable information for readers. The objectives and rationale are clearly outlined and structured. However, certain points need clarification, and additional information would enhance understanding.

We thank the reviewer for the kind and useful comments and suggestions to improve the document.

MainComments:

  1. L. 14-26:Some data and important results should be added in abstract.

More data and results have been included in the abstract.

  1. L. 77: It is recommended to describe how the control group should be prepared.

To make it clearer, “(plain formulation)” has been added.

  1. L. 109: "pH" revised to "pHvalue".

"pH" has been replaced with "the pH value".

  1. L. 130: "…S. aureus…", please spell the full words in fist time mention.
  2. L. 131: "…L. monocytogenes…", please spell the full words in fist time mention.

They are now mentioned before, with their full words.

  1. L. 150, 152: "7°C, 200°C" revised to "7°C, 200 °C".

The mistakes have been amended.

  1. L. 176, 207, 293, 337, 382, 427, 518:"Tukey test" revised to "Tukey's test".

“Tukey test” has been replaced with “Tukey’s test”.

  1. L. 202: "…and pH were…" revised to "…and pH value were…".

"…aw and pH were…" has been replaced with "…and the aw and pH values were…".

  1. L. 206, 425: Please check the correctness of the statistical data and labeling in Table1 & 5.

The statistical data has been checked and seem correct to us. “raw” has been replaced with “row”.     

  1. L. 244: Please confirm “Table1orTable 3”?

We apologize for the mistake, which has been amended.

  1. L. 526,669,718:Please uniform the reference mark.

The mistakes have been amended.

  1. Please explain the basis for the concentration of white grape pomace(0.5%Pomace & 3%Pomace)in the research.

The following text has been added (section 2): “The pomace percentages were chosen on the basis of previous work in burgers showing that 0.5% was the lower percentage with any beneficial effect, whereas 3% was the higher percentage not causing panelists’ rejection”.

  1. Table 1. Please adjust the table data format to make it easier for readers to understand.

It has been amended.

  1. Table 5.It is recommended that all data in this table be statistically analyzed so that the complete research results can be more easily compared and explained.

All the data were subjected to an ANOVA (a common statistical test), and the results are included in the last column (labeled as “P”). When the ANOVA was significant at the P ≤ 0.05 level, the Tukey test was performed, as it is common practice and as it was indicated in old lines 175-176. When the Tukey test was significant, different letters were included in the same row, as it is indicated in old line 427. When it was not significant (which can happen because the Tukey test is less statistically powerful than the ANOVA), no letters were included. This is a usual way of expressing this type of statistical results.

Reviewer 3 Report

Comments and Suggestions for Authors

This study was performed to determine the replacement effect of nitrite with HPP-stabilized white grape pomace on the quality attributes of dry-cured sausages. The experimental design and approaches are reasonable, but some issues should be handled for further publication.

Title: The purpose of adding the white grape pomace should be specified in the title. An alternative source for which additives?

L46 This reviewer does not agree that such substitutes can replace all synthetic additives used in meat products. Should be specified.

L72 The muscle name of the used lean pork should be described.

L76 The mixing condition should be described (vacuum condition, rotation speed, and temperature).

L91 What kind of natural casing was used for?

L160-171 The method of pre-session should be mentioned.

L214-217 This speculation is quite complicated since there was no pH drop in 0.5%Pomace. It should be rephrased.

L218-219 This reviewer disagrees with this suggestion since all samples were processed within the same days. Moreover, there was a numerically small difference with zero standard deviation. That is just an unverified guess.

L245 “Some compounds” should be specified.

L416-417 Color defect is one of the most influential factors affecting consumer preferences. This tendency could also be confirmed in the results of sensory analysis (Table 7). Thus, the solving strategy should be discussed.

In Table 1, the post-hoc result of moisture content between treatment should be addressed since the significance of one-way ANOVA was observed.

In Table 7, appropriate term should be added to the top of the column representing the characteristics.

Author Response

Answers to the comments and suggestions

This study was performed to determine the replacement effect of nitrite with HPP-stabilized white grape pomace on the quality attributes of dry-cured sausages. The experimental design and approaches are reasonable, but some issues should be handled for further publication.

We appreciate the relevant comments and suggestions to improve the document.

Title: The purpose of adding the white grape pomace should be specified in the title. An alternative source for which additives?

We appreciate the comment. We have modified the title to fit better the content of the document.

L46 This reviewer does not agree that such substitutes can replace all synthetic additives used in meat products. Should be specified.

The paragraph has been reorganized and the sentence rewritten for clarity.

L72 The muscle name of the used lean pork should be described.

The required information has been included: “whole deboned leg pork”.

L76 The mixing condition should be described (vacuum condition, rotation speed, and temperature).

The required information has been included. The mixing machine does not allow temperature control and, therefore, we have included the initial meat temperature.

L91 What kind of natural casing was used for?

The required information has been included: “dry pork…previously soaked”.

L160-171 The method of pre-session should be mentioned.

Some information on the panel and training prior to the sessions has been included.

L214-217 This speculation is quite complicated since there was no pH drop in 0.5%Pomace. It should be rephrased.

The sentence has been rewritten for clarity.

L218-219 This reviewer disagrees with this suggestion since all samples were processed within the same days. Moreover, there was a numerically small difference with zero standard deviation. That is just an unverified guess.

The sentence has been rewritten for clarity.

L245 “Some compounds” should be specified.

For clarity, “…some compounds…” has been replaced with “…a fraction of the compounds quantified as phenols…”

L416-417 Color defect is one of the most influential factors affecting consumer preferences. This tendency could also be confirmed in the results of sensory analysis (Table 7). Thus, the solving strategy should be discussed.

This point has been discussed now after that sentence.

 In Table 1, the post-hoc result of moisture content between treatment should be addressed since the significance of one-way ANOVA was observed.

The post-hoc test for moisture showed no significant differences (the table footprint indicates that letters are only included when there are significant differences), which can occur because the Tukey test is less statistically powerful than the ANOVA.

Some information has been added to old line 214 to clarify this point: “…(although the Tukey test was not sufficiently powerful to reveal significant differences between groups).”

In Table 7, appropriate term should be added to the top of the column representing the characteristics.

We thank the comment. There is not any term since the table title indicates what was included in the table, as it was done with the other tables. Of course, this information could be included again in all the tables if needed.

Reviewer 4 Report

Comments and Suggestions for Authors

Manuscript 2857522

Journal Foods

Title HHP-Stabilized White Grape Pomace as an Alternative for Additives In Dry-Cured Sausages (“Salchichón”)

The manuscript entitled “HHP-Stabilized White Grape Pomace as an Alternative for Additives In Dry-Cured Sausages (“Salchichón”)” describes the use of white grape pomace as additive of dry-Cured sausages and its effect on lipid oxidation, protein oxidation, colour, volatile compounds, sensory characteristics, and the microbiological status of the product. The topic is not original and novel. Antioxidant activity of sausages is not reported and a general re-organization and improvement is necessary. Please follow the comments in the file.

Comments on the Quality of English Language

Several sentences need improvement

Author Response

Answers to the comments and suggestions

The manuscript entitled “HHP-Stabilized White Grape Pomace as an Alternative for Additives In Dry-Cured Sausages (“Salchichón”)” describes the use of white grape pomace as additive of dry-Cured sausages and its effect on lipid oxidation, protein oxidation, colour, volatile compounds, sensory characteristics, and the microbiological status of the product. The topic is not original and novel. Antioxidant activity of sausages is not reported and a general re-organization and improvement is necessary. Please follow the comments below:

We appreciate the help to improve the document.

We would kindly point out that our study provides new information on the effect of adding whole grape pomace on the characteristics (general, instrumental color, oxidation, volatile compounds, sensory traits) of dry-cured sausages, with an improvement on their oxidation stability. Previous studies using grape pomace-based extracts, but not whole pomace, have been performed. Our pomace, stabilized by blanching and applying high hydrostatic pressure and without drying and/or solvent extraction, is a green alternative that overcomes some potential consumers’ concerns about the use of solvents and chemicals.

Regarding the antioxidant activity, we think that it is especially relevant for the grape pomace (where it was determined), whereas for the dry-cured sausages the most relevant parameters related to oxidation are lipid and protein oxidation. We will consider the suggestion for future studies.

The suggestions for the reorganization and improvement have been attended.

We appreciate the relevant comments and suggestions to improve the document.

L14-27 Revise the abstract. Please add quantitative data

We thank the comment. More data have been included.

L35-38 Please be more specific. Which are these attempts? Which are the natural products? Which is the effect of their addition on meat quality and safety?

Some information on those natural products and their effect has been included.

L47-54 Please add more details related to this part. Which is the effect of grape pomace addition on the quality and safety of the product, the physico-chemical characteristics, lipid oxidation, oxidative stability, colour, and sensory properties? Add more data

More information has been included, and it is now mentioned that some studies have not researched the effect on the microorganisms.

L62 Why authors proposed the stabilization by using HHP? Which are the advantages compared to drying? Please be more specific. It is not clear in the text. Revise the text

More information on drying has been included, and the text has been revised.

L65 Please add more details on your previous work. It is important to support the objective of this work

More information on our previous has been included.

L82-89 Please move this part in a new section named “Production and stabilization of white grape pomace”. More details are necessary on the production and characteristics of white grape pomaces, albeit already described in Martín-Mateos et al. (2023).

The text has been moved to the new section and more information has been included.

L77-81 How the authors selected the grape pomace concentration added to sausages? Please explain in the text

The following text has been added: “The pomace percentages were chosen on the basis of previous work in burgers showing that 0.5% was the lower percentage with any beneficial effect, whereas 3% was the higher percentage not causing panelists’ rejection”.

L97-105 Why the antioxidant activity was measured only on grape pomace? Why the authors did not evaluate the antioxidant activity of sausages? Please explain. The antioxidant activity of the product could be included in the manuscript

We think that the antioxidant activity is relevant for the grape pomace, whereas for the dry-cured sausages the most relevant parameters related to oxidation are lipid and protein oxidation. We will consider the suggestion for future studies.

L106-111 Please better describe the determination of protein content and fat.

The references for the methods have been included and the determinations are better described.

L96-120 Please add the calibration curve used and the concentrations of standard for each assay

The details have been included.

L121-131 Please add the media and incubation conditions for all the microbial groups

The details are now included.

L145 37°C…how this temperature was selected? Please explain in the text

This is a temperature commonly using in SPME-GC-MS analysis, and within the usual range for meat products. Some information is now included in the text. More information and references could be included if needed.

L160 Trained panelists…How panelists were trained? Please be more specific

Some information has been included. More information, and references, could be included if needed.

L160-164 Please explain the test used and the significance of the data reported in Table 7 (e.g., 6.9 is better than 4.7?). Rewrite this part adding significant information.

More information has been included to make clear that the descriptive method measured the intensity of 12 traits. Descriptive methods are commonly used for food analysis. An intensity measure does not indicate directly whether a higher value is better than a lower one, except for traits such as defects, for which a lower value is better. THe discussion now includes some information on this topic. More information and references for descriptive methods and/or scale measurement could be included if required.

L179-180 Move this sentence in Results

The sentence has been moved to section 3.6.

L192-104 Please move this part in the section 3.1

The sentence in lines 192‒194 has been rewritten and moved to section 3.1.

The sentence in lines 194‒195 has been moved to section 2.1.

L198-200 Rewrite It is not correct in English

The sentence has been rewritten for clarity.

L263-266 Please explain this sentence. It is not clear

The text has been rewritten for clarity.

L271-274 It is not sufficient to exclude the determination of the antioxidant activity of sausages. Please explain

The text has been rewritten for clarity.

L300-302 LAB counts are equally to the control. Delete or revise this sentence

The text has been rewritten and a sentence added for clarity.

L305-310 Please expand the discussion of this part including the mechanism of action of polyphenols against bacteria and the factors affecting the antimicrobial activity of polyphenols. The paper doi.org/10.3390/foods12122315 is suggested for your analysis and discussion

The text has been rewritten for clarity. We have tried to keep the new text as concise as possible (five new lines) to avoid including extensive information that can be found in the two references now included here. More references for reviews and research articles, as well as more text, could be included if required.

We thank the kind suggestion for references to improve the document. However, we are afraid that the document already has a large number of them and we have had to add some more during the revision and, therefore, we have included here the references of two review papers already mentioned in the introduction section for the antimicrobial activity [6,10].

L313-315 Please add a discussion on the antibacterial activity of white grape pomace extracts. Which are the Minimum Inhibitory Concentration values? Which are the compounds responsible for the antibacterial effect? The paper doi.org/10.3390/molecules26195918 is suggested for your analysis and discussion

The text has been rewritten for clarity. The sentences just summarize the results for the HPP-stabilized pomace and, therefore, we think there is no need to expand the discussion here.

We thank the kind suggestion for references to improve the document. However, we are afraid that the document already has a large number of them and, therefore, we prefer to maintain here the references of two review papers mentioned in the introduction section for the antimicrobial activity [6, 10].

L329-332 Rewrite. It is not correct in English

The text has been rewritten for clarity.

Table 3 Add the statistical analysis related to protein oxidation

The results were already included in Table 3: P = 0.345. The Tukey test was only performed when a significant effect was found in the ANOVA, which is common practice, and this was indicated in old lines 175‒176.

L348-352 The antioxidant activity of sausages is not reported. It is a limitation of this study.

The lines just deals with the lipid oxidation (TBARS) results. We will consider measuring the antioxidant activity of the sausages in future studies.

L366-369 Which is the reason? Please add a possible explanation

The required discussion was included in lines 357‒359. More discussion on the topic has been added.

L394-398 Please better discuss these results

The text has been rewritten for clarity.

L407-415 Please better explain this part. How the colour is correlated with the microbial counts in sausages? Please explain in the text with a brief introduction

A brief introduction and a reference have been included and the text has been rewritten for clarity.

L407-412 Here and throughout the manuscript. Please describe the correlation analysis in Materials and Methods. Now it is not described but reported only in Results and Discussion.

The correlation analysis was described in old lines 183‒184 (Materials and Methods section).

L433-446 Rewrite this part highlighting the different trends (e.g., compound X is higher in sample Y than sample Z rather than “generic differences between groups”)

We apologize for the repeated sentences in the second paragraph, which has been removed. Now the first paragraph enumerates the compounds significant different. Then, the detailed information and discussion on each compound is presented, starting with the compounds with significant differences that were identified in the pomace, and finishing with the ones not identified in the pomace (pages 12 and 13).

L476-478 Which is the reason? Please expand the discussion on this aspect

A reason is now included and the discussion has been expanded.

L492-493 Rewrite. It is not clear

The text has been rewritten for clarity.

Table 6 Why 3-hexen-1-ol, 2-hexen-1-ol, or 3-hexan-1-ol were not detected in sausages? Please discuss this aspect in the text

Some discussion on the topic has been included just before Table 5.

L505 Please cite Aquilani et al. with the reference’ number.

The mistake has been amended.

L531-537 Please discuss these results. For example, why nitrasc samples showed a different lean and fat colour?

More discussion has been included. More discussion on the topic was included in old lines 554‒561.

L554-577 I would suggest to place the PCA results in a separate section and to discuss all the properties of the sausages and their correlation

We thank the suggestion. However, the PCA and the correlationships included in this section were performed to help the discussion of the results included in Table 7 and, therefore, we think it should not be separated.

L595 delete without adding synthetic additives. Grape pomace addition has detrimental effects on colour and no data is reported on the safety in case of pathogen contamination. Therefore, a complete substitution of additives should be not suggested. Revise here and in the conclusion section.

“without adding synthetic additives” in old line 595 has been replaced with “by adding a grape by-product processed without chemicals”. It has been removed from the Conclusions section. The complete substitution is not suggested.

Round 2

Reviewer 4 Report

Comments and Suggestions for Authors

Authors addressed reviewer's comments. The paper is acceptable for publication.

Comments on the Quality of English Language

Minor editing changes are necessary